# Imaging-Guided Evaluation of the Novel Small-Molecule Benzosuberene Tubulin-Binding Agent KGP265 as a Potential Therapeutic Agent for Cancer Treatment

**DOI:** 10.3390/cancers13194769

**Published:** 2021-09-24

**Authors:** Yihang Guo, Honghong Wang, Jeni L. Gerberich, Samuel O. Odutola, Amanda K. Charlton-Sevcik, Maoping Li, Rajendra P. Tanpure, Justin K. Tidmore, Mary Lynn Trawick, Kevin G. Pinney, Ralph P. Mason, Li Liu

**Affiliations:** 1Department of Radiology, University of Texas Southwestern Medical Center, Dallas, TX 75390, USA; guoyihang@csu.edu.cn (Y.G.); honghong2021@tjh.tjmu.edu.cn (H.W.); jeni@gerberich.us (J.L.G.); limaoping@hospital.cqmu.edu.cn (M.L.); 2Department of Gastrointestinal Surgery, The Third XiangYa Hospital of Central South University, Changsha 410013, China; 3Department of Medical Ultrasound, Tongji Hospital, Tongji Medical College, Huazhong University of Science and Technology, Wuhan 430074, China; 4Department of Chemistry and Biochemistry, Baylor University, Waco, TX 76798, USA; Samuel_Odutola@alumni.Baylor.edu (S.O.O.); Amanda_Charlton@baylor.edu (A.K.C.-S.); Rajendra_Tanpure@alumni.baylor.edu (R.P.T.); Justin_Tidmore@alumni.baylor.edu (J.K.T.); Mary_Lynn_Trawick@baylor.edu (M.L.T.); Kevin_Pinney@baylor.edu (K.G.P.); 5Department of Ultrasound, The First Affiliated Hospital of Chongqing Medical University, Chongqing 400016, China; 6Simmons Cancer Center, University of Texas Southwestern Medical Center, Dallas, TX 75390, USA

**Keywords:** vascular-disrupting agent (VDA), bioluminescence imaging (BLI), benzosuberene, combretastatin, carboplatin, breast tumors, kidney tumors, multispectral optoacoustic tomography (MSOT), photoacoustics

## Abstract

**Simple Summary:**

Vascular-disrupting agents promise significant therapeutic efficacy against solid tumors by selectively damaging tumor-associated vasculature. Dynamic BLI and oxygen-enhanced multispectral optoacoustic tomography (OE-MSOT) were used to compare vascular shutdown following administration of KGP265. BLI signal and vascular oxygenation response (ΔsO_2_) to a gas breathing challenge were both significantly reduced within 2 h indicating vascular disruption, which continued over 24 h. Twice-weekly doses of KGP265 caused a significant growth delay in MDA-MB-231 human breast tumor xenografts and 4T1 syngeneic breast tumors growing orthotopically in mice.

**Abstract:**

The selective disruption of tumor-associated vasculature represents an attractive therapeutic approach. We have undertaken the first in vivo evaluation of KGP265, a water-soluble prodrug of a benzosuberene-based tubulin-binding agent, and found promising vascular-disrupting activity in three distinct tumor types. Dose escalation in orthotopic MDA-MB-231-luc breast tumor xenografts in mice indicated that higher doses produced more effective vascular shutdown, as revealed by dynamic bioluminescence imaging (BLI). In syngeneic orthotopic 4T1-luc breast and RENCA-luc kidney tumors, dynamic BLI and oxygen enhanced multispectral optoacoustic tomography (OE-MSOT) were used to compare vascular shutdown following the administration of KGP265 (7.5 mg/kg). The BLI signal and vascular oxygenation response (ΔsO_2_) to a gas breathing challenge were both significantly reduced within 2 h, indicating vascular disruption, which continued over 24 h. A correlative histology confirmed increased necrosis and hemorrhage. Twice-weekly doses of KGP265 caused significant growth delay in both MDA-MB-231 and 4T1 breast tumors, with no obvious systemic toxicity. A combination with carboplatin produced significantly greater tumor growth delay than carboplatin alone, though significant carboplatin-associated toxicity was observed (whole-body weight loss). KGP265 was found to be effective at low concentrations, generating long-term vascular shutdown and tumor growth delay, thus providing strong rationale for further development, particularly in combination therapies.

## 1. Introduction

Solid tumor growth beyond about 1–3 mm in diameter depends extensively on angiogenesis initiating neovascularization for the supply of nutrients and oxygen [1]. However, tumor neovasculature is abnormal, in terms of both structure and function, and has been proposed as a specific target for chemotherapy [2,3,4,5]. Notably, tumor endothelial cells undergo rapid proliferation, and blood vessels are ill-formed, leaky and generally lack pericyte coverage [6,7]. Two types of therapy have been proposed to target tumor-associated vasculature: antiangiogenic agents inhibit the development of blood vessels a priori [8], while vascular-disrupting agents (VDAs) specifically target the existing neovasculature [4,5,9,10,11]. Many small-molecule VDAs interact with the tubulin-microtubule protein system at the well-characterized vinca alkaloid and colchicine sites, which are located separately on the αβ-tubulin heterodimer [12]. Colchicine and vinblastine cause microtubule destabilization, leading to vascular disruption, but at concentrations that are close to their maximum tolerated dose, mandating a need for modified agents [13,14].

In the late 1970s, Pettit and coworkers discovered the combretastatin series of natural products in the South African bush willow tree, *C**ombretum caffrum*, of which combretastatin A-1 (CA1) [15] and combretastatin A-4 (CA4) [16] are two of the most potent compounds, each exhibiting pronounced biological activity as inhibitors of tubulin polymerization and selective vasculature disrupting agents. VDA activity results from microtubule disruption in activated endothelial cells, which initiates a signaling pathway characterized by profound cytoskeletal and morphology changes [17,18]. Consequently, endothelial cells round up, leading to enhanced vascular leakage and detachment from each other and from the underlying substratum, causing vascular congestion [19]. Direct vascular disruption is predicted to cause massive downstream starvation and hypoxiation, thereby potentiating the local effect and generating extensive necrosis [4].

Although VDAs were proposed originally some 30 years ago [3], there remains much interest in developing effective translational agents. Several VDAs have been evaluated in clinical trials, though, to date, none has received FDA approval [20,21]. CA4P (prodrug of CA4) was granted the status of an orphan drug by the European Medicines Agency (EMA) [20]. It is recognized that VDAs are ineffective as monotherapies, since a thin peripheral rim of cells, thought to receive nutrients from the host vasculature, survives, even after destruction of the tumor vasculature. While the tumor center may necrose, the rim often repopulates rapidly. As such, several VDAs have been tested in combination with additional therapies, including radiotherapy [22,23], antiangiogenic agents (such as bevacizumab) [24] and, recently, immunotherapy [25]. There is a current resurgence of interest in VDAs, and frequent reports describe novel agents, many based on the combretastatin motif [26,27,28,29,30,31,32,33,34,35,36], including benzosuberenes [37,38,39,40]. 2-Methoxy-5-(3′,4′,5′-trimethoxyphenyl)-6,7,8,9-tetrahydro-5H-benzocyclohepten-1-ol (also referred to as KGP18; Figure 1) has structural similarities to the combretastatins (e.g., the trimethoxyaryl group) and was found to exhibit exceptional biological properties, suggesting it would provide effective tumor vascular disruption. Notably, KGP18 demonstrated a potent inhibition of tubulin polymerization (IC_50_ = 1.7 µM) and was strongly cytotoxic, as evidenced by the growth inhibition (GI_50_ < 50 pM) of several tumor cell lines [37]. In common with the combretastatins (CA1 and CA4), KGP18 proved to be poorly water-soluble, and as for the combretastatins, we found that the phosphorylation of the free phenolic group yielded a water-soluble prodrug (KGP265; Figure 1), which still provided effective cell growth inhibition (GI_50_ ~9 nM) [38]. As expected, this was higher than for the free drug due to the need for the enzymatic release of KGP18. The proposed mechanism of KGP265 is presented in Figure 1.

In developing any new drug, noninvasive assays of efficacy provide vital insight into their activity with respect to the dosing and pharmacodynamics of action. Many noninvasive imaging methods are available [41], but bioluminescence imaging (BLI) has found an increasingly important role in small animal research for monitoring tumor growths using tumor cells transfected to express luciferase. Noting that BLI depends on the delivery of the substrate luciferin to the tumor, we have previously demonstrated the utility of dynamic bioluminescence imaging (BLI) to evaluate several VDAs, including CA4P and analogs [5,40,42,43,44], and this approach has also been used by others [45]. We have now applied this approach to examine the efficacy of KGP265 in an orthotopic MDA-MB-231 breast mammary fat pad tumor model in mice. The MDA-MB-231 breast tumor has been widely reported in previous investigations of vascular-disrupting activity in vivo [46,47]. To further demonstrate its efficacy, we compared the activity of KGP265 in other well-characterized tumors types, specifically syngeneic orthotopic mouse breast 4T1 [48] and RENCA kidney tumors [49]. Many other imaging approaches have been applied to evaluate the activity of VDAs [50], and we have further characterized the efficacy of KGP265 using multispectral optoacoustic tomography (MSOT). Recently, the available MSOT has been particularly attractive, since it provides specific measurements of oxy- and deoxyhemoglobin, yielding insights into hemoglobin concentrations and oxygen saturation (sO_2_) noninvasively, and avoids the need for luciferase transfection. It has been reported that the change in sO_2_ in response to an oxygen gas breathing challenge is more reliable in terms of characterizing the tumor vasculature than a simple static baseline measurement [51,52,53], and we have now applied oxygen-enhanced MSOT to further characterize the activity of KGP265.

It is anticipated that VDAs will be applied clinically in combination therapy, and we have therefore compared the tumor growth delay generated by combination with the traditional cytotoxic chemotherapeutic agent carboplatin in MDA-MB-231 human breast tumor xenografts and syngeneic 4T1 breast tumors growing orthotopically in mice.

## 2. Materials and Methods

### 2.1. Cell Studies

#### 2.1.1. Cell Cycle Analysis

The cell cycle distribution was evaluated by flow cytometry. Briefly, adherent MDA-MB-231 cells in 6-well culture plates were treated with KGP18 or KGP265 (0.01, 0.1, 0.5 and 1 µM) for 48 h. Both adherent and nonadherent cells were collected, washed twice with ice-cold PBS and fixed in 70% ethanol overnight at 4 °C. Cell pellets were suspended in PBS containing RNase (10 mg/mL) and stained with propidium iodide (6.7 mg/mL) at 4 °C in the dark for 2 h. The DNA content was determined based on propidium iodide fluorescence determined with a BD FACsCalibur (San Jose, CA, USA) [18].

#### 2.1.2. Endothelial Tube Disruption Assay

Human umbilical vein endothelial cells (HUVECs) were plated on Matrigel-coated (9.5 mg/mL, BD Pharmingen, San Diego, CA, USA) 24-well culture plates (Corning Life Sciences, Glendale, AZ, USA) at a concentration of 124,000 cells/well and maintained at 37 °C for 16 h in M200 medium supplemented with a high growth factor supplement kit (ATCC). Tube disruption was induced by treatments with varying concentrations of KGP18 for 2 h; after which, the medium was removed, and the cells were washed twice with fresh M200. The cells were imaged (nine fields per well) using an Axiovert 40 CFL inverted microscope (Zeiss, Thornwood, NY, USA) at 5× magnification, and the bright field images were collected with negative contrast using a Canon Powershot A640 digital camera mounted onto the microscope [18].

#### 2.1.3. Fluorescence Imaging of Endothelial Cells

Actively proliferating HUVECs were plated at 10,000 cells/glass coverslip coated with 1% gelatin in 6-well culture plates (Corning Life Sciences, Glendale, AZ, USA) using high growth factor-supplemented medium and incubated at 37 °C for 48 h to approximately 40% confluence before beginning treatment. The cells were treated with KGP18 (2 nM and 20 nM) from stock solutions made in DMSO (final concentration of DMSO less than 0.5% in the media) for 2 h. After treatment, the cells were fixed and permeabilized with a solution of 4% paraformaldehyde and 0.5% Triton X (Sigma-Aldrich, St. Louis, MO, USA) in PBS. Microtubules were detected with mouse anti-α-tubulin antibody (Sigma-Aldrich), followed by incubation with FITC-conjugated goat anti-mouse IgG (Jackson Immuno research, West Grove, PA, USA), actin fibers were stained using Texas Red-conjugated phalloidin (Invitrogen, Waltham, MA USA) and nuclear staining was carried out with 4’,6-diamidino-2-phenylindole (DAPI). The fluorescence and phase contrast images were collected using an Olympus FV 1000 confocal microscope with Olympus fluoview software (Olympus Imaging America Inc., Center Valley, PA, USA) using a 60× oil immersion objective [18].

#### 2.1.4. Alkaline Phosphatase (AP) Cleavage Assay

The enzymatic activity of alkaline phosphatase (from a human placenta, Sigma-Aldrich, St. Louis, MO, USA) was determined by using 4-nitrophenyl phosphate (Sigma-Aldrich, St. Louis, MO, USA) as the substrate; one unit hydrolyzed one micromole of 4-nitrophenyl phosphate per minute at 37 °C. The cleavage of prodrug KGP265 (100 µM) by AP (1 unit) was carried out at 37 °C in 10 mM glycine buffer solution (pH 8.6, containing 2 mM MgCl_2_) over 24 h. The release of KGP18 (retention time 7.3 min) was monitored by HPLC (Agilent Technologies 1200 series, with a ZORBAX Eclipse XDB-18 column) with a mobile phase of 1:1 acetonitrile/water (containing 0.05% TFA) at 1 mL/min. The control reaction consisted of KGP265 (100 µM) incubated for 24 h in a pH 8.6 glycine buffer solution without AP at 37 °C.

#### 2.1.5. Cell Culture

The human breast cancer cell line MDA-MB-231-luc and mouse breast cancer cell line 4T1-luc were gifts from Stanford University, and the mouse kidney cancer cells RENCA-luc were from UTSW. These cells were maintained in RPMI medium, 1% penicillin and streptomycin and supplemented with 10% heat-inactivated FBS. The cell lines were tested for mycoplasma using a MycoFluor™ Mycoplasma Detection Kit (Molecular Probes, Eugene, OR, USA), and the results were negative.

### 2.2. Mouse Models

MDA-MB-231-luc tumors were induced in the left upper mammary fat pad (MFP) initially of nude and, later, NOD/SCID mice aged 6–8 weeks. 4T1-luc tumors were induced in the right lower MFP of syngeneic BALB/c mice aged 6–8 weeks. Briefly, the mice were anesthetized by exposure to 1–3% isoflurane and injected with 1 × 10^6^ cells in a volume of 50–100 μL suspended in a mixture of 75% culture medium/25% Matrigel (BD Pharmingen, San Diego, CA, USA) into the mammary fat pad of female mice. RENCA-luc orthotopic tumors were induced by the injection of cells (1 × 10^6^) into the right kidney of syngeneic BALB/c mice (age 8 weeks from the UTSW breeding core) [54]. Tumor growth was monitored weekly by BLI, and the external caliper measurements (tumor size = [length × width × height] × π/6), and investigations were initiated when the tumors reached a size of 5–8 mm.

### 2.3. Bioluminescence Imaging (BLI)

Anesthesia was induced with 1–3% isoflurane, and the mice were then placed on the warmed imaging stage of an IVIS Spectrum^®^ small animal imaging system (PerkinElmer Inc., Waltham, MA, USA) with continuous exposure to 1 to 2% isoflurane, as described previously [43]. Anesthetized mice were placed at similar orientations for successive images to ensure consistent signal detection. For in vivo imaging, the mice were administered substrate *D*-luciferin (sodium salt; Gold Biotechnology, St. Louis, MO, USA) by subcutaneous injection (80 µL) in the foreback neck region with a 40 mg/mL solution in 0.9% saline. Generally, 2–5 mice were imaged simultaneously, for both weekly assessments of tumor growth, as well as for dynamic BLI studies. Grey-scale photographs were acquired. The emitted light was detected using pairs of images, one set to automatic exposure and the other to 1 s, to ensure that intense and weak signals were effectively acquired. Images were acquired every minute over the course of 35 min following luciferin injection. The quantitative data, such as the photon flux, were automatically normalized for camera aperture, camera-to-subject distance and exposure time. For each mouse, a region of interest (ROI) was selected, which was readily identified based on the appearance of the tumor in the photograph and, more importantly, the emitted light. Since there was essentially no background signal, the contours could extend beyond the tumor without significantly altering the measured light intensity.

The BLI signal was quantified as the total flux in photons/second using Living Image Software 4.2 (http://www.perkinelmer.com, accessed on 20 September 2019). The time and maximum intensity of the integrated BLI signal were identified for the baseline measurement (pretreatment). To assess the changes in tumor perfusion, the BLI signal was subsequently determined at the same time point following the fresh luciferin administration. Imaging was repeated with the administration of fresh luciferin at 2, 4 and 24 h after drug administration.

### 2.4. Multispectral Optoacoustic Tomography (MSOT)

4T1-luc and RENCA-luc-implanted BALB/c mice were imaged at baseline, 2 and 24 h after the administration of KGP265 or saline (control). Mice were shaved around the tumor region and residual hair removed with depilatory cream (Nair, Church & Dwight, Ewing, NJ, USA). Mice were anesthetized via the inhalation of 2% isoflurane in the air in an induction chamber and then transferred to the animal holder. A thin layer of ultrasound gel (Aquasonic Clear, Parker Labs, Fairfield, NJ, USA) was applied to the surface of the mouse, particularly around the tumor region, to ensure effective optical and acoustic contact to the polyethylene membrane. To examine oxy- and deoxyhemoglobin, mouse images were acquired in transaxial sections through the tumor region using seven wavelengths: 680, 715, 730, 768, 800, 850 and 900 nm with an MSOT InVision 256-TF device (iThera Medical, Munich, Germany), as described previously [53]. Briefly, a tunable optical parametric oscillator (OPO) pumped by an Nd:YAG laser provided excitation pulses with a duration of 9 ns at wavelengths from 680 nm to 980 nm at a repetition rate of 10 Hz with a wavelength tuning speed of 10 ms and a peak pulse energy of 90 mJ at 720 nm. Seven arms of a fiber bundle provided the uniform illumination of a ring-shaped light strip of approximately 8 mm in width. For ultrasound detection, 256 toroidally focused ultrasound transducers with a center frequency of 5 MHz (60% bandwidth), organized in a concave array of 270 degrees of angular coverage and a radius of curvature of 4 cm, were used. A model-based reconstruction was used prior to multispectral processing. Ten frames per wavelength were acquired and averaged. The animal was placed in the imaging chamber (34 °C) for ten minutes to reach thermal equilibrium before imaging. Initially, the tumor region was imaged while breathing air to select one slice with the biggest tumor diameter. After two minutes of imaging with air, the inspired gas was changed to 100% oxygen. A single slice approximately in the middle of the tumor with the largest cross-section was imaged continuously during the dynamic gas challenge over a period of five minutes to allow the animal to reach equilibrium with the new gas.

Image reconstruction and data analysis were carried out with ViewMSOT (Version 3.8 iThera Medical, GmbH, Munich, Germany), a software that covers the full imaging chain: from data acquisition to image reconstruction, spectral unmixing, visualization and quantification. Images were reconstructed with a pixel size of 75 µm × 75 µm. We performed the analysis of the sO_2_ map with MATLAB 2016b (MathWorks) using custom software and visualized the images with ImageJ (Release 1.52a, National Institutes Health, USA). ROIs were drawn around the entire tumor region from the largest cross-section of the tumor and a healthy, well-vascularized tissue region around the spine. No fluence correction was applied. The percentage of hemoglobin saturation (%sO_2_) was calculated as:(1)$${\mathrm{sO}}_{2}=\frac{\left[{\mathrm{HbO}}_{2}\right]}{\left[{\mathrm{HbO}}_{2}\right]+\left[\mathrm{Hb}\right]}\times 100\%.$$

A negative value in either oxy- or deoxyhemoglobin resulted in a NaN value and was discarded for that region. Black pixels also represented areas lacking a detectable hemoglobin concentration. The average sO_2_^MSOT^ was calculated pixelwise in each ROI for the air and oxygen breathing periods and are presented as sO_2_^MSOT^ (Air) and sO_2_^MSOT^ (O_2_), respectively. The amplitude of the response to the oxygen gas challenge
ΔsO_2_^MSOT^ = sO_2_^MSOT^ (O_2_) − sO_2_^MSOT^ (Air)(2)
was calculated for each pixel. The change in response following KGP265 treatment was calculated as:ΔΔsO_2_^MSOT^ = ΔsO_2_^MSOT^ (2 or 24 h) − ΔsO_2_^MSOT^ (baseline),(3)
as shown for each tumor type with respect to the intervention.

### 2.5. Drug Dosing and Preparation

KGP265 was synthesized in the Pinney laboratory using published procedures [38]. For the in vivo experiments, KGP265 was dissolved in saline at various concentrations and dosed at 5–30 mg/kg in a total injected volume of 100 μL IP (*n* = 5 or 6 mice per dose). Dynamic BLI images were acquired before treatment to assess the baseline signal and up to 24 h after treatment.

### 2.6. Immunohistochemistry

Coinciding with each imaging time point, pimonidazole (Hypoxyprobe Plus Kit; Hypoxyprobe Inc., Burlington, MA, USA) was used to reveal hypoxia in select mice. Thirty minutes to one hour before sacrifice, the animals were injected IV with 60 mg/kg pimonidazole according to the manufacturer’s directions. The tissues were fixed in 4% paraformaldehyde overnight and then processed and embedded in paraffin. The sections were cut at 5 µm, and routine H&E staining was performed. For immunohistochemistry staining, the slides were deparaffinized in Clear-Rite 3 (Thermo Fisher Scientific, Waltham, MA, USA) and rehydrated in a series of different concentrations of ethanol. Antigen retrieval was performed with a citrate buffer of pH 6.0 (Sigma-Aldrich) for 20 min in a 98 °C water bath; then, the slides were washed 3 times in Tris-buffered saline (TBS) (Thermo Fisher Scientific). Endogenous peroxidase was quenched with 3% H_2_O_2_ in water for 10 min at room temperature. After washing 3 times in TBS, the slides were blocked with 1% bovine serum albumin (BSA) in TBS for 30 min at room temperature. The sections were incubated with a 1:100 dilution of FITC-Mab1 in BSA/TBS for 60 min at room temperature, then washed 3 times with TBS. The slides were incubated with a 1:100 dilution of Rab-α-FITC-HRP for 30 min at room temperature, then washed (3 times) with TBS. DAB chromagen (Vector ImmPACT DAB substrate kit, Vector Laboratories, Burlingame, CA, USA) was applied to the sections for 5 min; then, the slides were placed in water. The slides were counterstained with Mayer’s Hematoxylin (Thermo Fisher Scientific) for 2 min. This same staining protocol was used for CD-31 using an overnight incubation of a 1:20 dilution of rat-α-mouse CD-31 (Dianova clone SZ31, Dianova, Hamburg, Germany) at 4 °C with the ImmPRESS HRP reagent kit: anti-Rat IgG, mouse-adsorbed (Vector Laboratories, cat no. MP-7444). For Ki-67, an overnight incubation of a 1:200 dilution RabpAb-α-Ki67 (ab15580, Abcam, Cambridge, MA) at 4 °C was performed, and the ImmPRESS HRP Anti-Rabbit IgG kit was used (Vector Laboratories, cat no. MP7451). The selected field in the tumor periphery was imaged using a Hamamatsu NanoZoomer 2.0HT microscope (Hamamatsu Photonics, Hamamatsu, Japan). NIH ImageJ software (National Institutes of Health, Bethesda, MD, USA) was used to quantify the degree of staining on the images by measuring the pixels, and the data were normalized to that of the corresponding control mice.

### 2.7. Drug Combination Therapy

Five cohorts of female SCID mice (UTSW breeding colony) were implanted with 1 × 10^6^ MDA-MB-231-luc cells, and four cohorts of female BALB/c mice (UTSW breeding colony) were implanted with 1 × 10^6^ 4T1-luc cells; in each case, the cells were mixed with 50% Matrigel™ (BD Biosciences, San Jose, CA, USA) and implanted in the left lower mammary fat pad. The treatment was initiated when the tumors reached approximately 300 mm^3^. The animals were divided into cohorts and treated by IP injection twice a week. MDA-MB-231-luc tumor cohorts: Group 1 (*n* = 13; mean volume 290 mm^3^) received saline (100 μL), Group 2 (*n* = 22; mean volume 310 mm^3^) received carboplatin (obtained from UTSW Pharmacy) administered at a dose of 50 mg/kg in a volume of approximately 100 μL, Group 3 (*n* = 7; mean volume 270 mm^3^) received KGP265 administered at a dose of 5 mg/kg, Group 4 (low-dose combination therapy; *n* = 8; mean volume 280 mm^3^) was treated with carboplatin (30 mg/kg), followed by KGP265 (3 mg/kg) two hours later and Group 5 (high-dose combination therapy; *n* = 8; mean volume 250 mm^3^) was treated with carboplatin (50 mg/kg), followed by KGP265 (5 mg/kg) two hours later. 4T1-luc tumor cohorts: Group 1 (*n* = 6; mean volume 280 mm^3^) received saline (100 μL), Group 2 (*n* = 8; mean volume 250 mm^3^) received carboplatin administered at a dose of 30 mg/kg in a volume of approximately 100 μL, Group 3 (*n* = 4; mean volume 260 mm^3^) received KGP265 administered at a dose of 3 mg/kg and Group 4 (low-dose combination therapy; *n* = 6; mean volume 230 mm^3^) was treated with carboplatin (30 mg/kg), followed by KGP265 (3 mg/kg) two hours later. The mice were weighed at least twice per week and terminated if they lost more than 20% body weight, showed severe toxicity or when the tumor reached 10% of its body weight. 

### 2.8. Statistical Analyses of Data 

Comparison of the BLI signal change was performed using a factorial ANOVA based on Fisher’s PLSD (protected least squares difference test) to compare multiple doses and times following treatments with respect to the percent signal loss normalized to the baseline for each tumor. The significance of the sO_2_^MSOT^ changes in the time series was assessed using a two-way unmatched ANOVA test (*p* < 0.05 was considered significant). Stained areas for immunohistochemistry were compared with unpaired *t*-tests. Drug efficacy was assessed based on tumor growth profile linear mixed models with first-order autoregression: AR (1) correlation structure, as well as using one way ANOVA, Kruskal-Wallis tests and pairwise Wilcoxon tests when appropriate. A survival analysis was performed using the Kaplan-Meier method and a log-rank test. For calculating the correlation coefficients, we used the Pearson formula for calculating the correlation coefficient *r* and *p*-value. Statistical analyses were performed using GraphPad Prism (version 8.3.0; GraphPad Software Inc., San Diego, CA, USA).

## 3. Results

### 3.1. KGP265 Mechanism and Cell Studies 

Nonspecific alkaline phosphatase is ubiquitously expressed in cells and tissues [55]. We have shown that the KGP265 prodrug (100 µM) was readily dephosphorylated (100%) in 24 h by alkaline phosphatase (1 unit) at pH 8.6 at a rate of 7.3 µM/min/unit enzyme activity to generate active KGP18, which functions as both a VDA and an antimitotic agent. No spontaneous hydrolysis of KGP265 was observed in the control reaction (no alkaline phosphatase). Both KGP18 and KGP265 demonstrated antimitotic effects via a blockade at the G2/M phase of the cell cycle in MDA-MB-231 breast cancer cells (Figure 2A). Endothelial cells can be induced to form a two-dimensional capillary network when plated on a model extracellular matrix (Matrigel) in the presence of appropriate growth factors. The KGP18 treatment of a pre-established HUVEC tube network resulted in cell rounding and clumping that disrupted the overall network (Figure 2B). Rapid and profound cytoskeletal changes were observed in activated endothelial cells (used to model the tumor endothelium) treated with 20 nM KGP18 (Figure 2C) that included the disruption of microtubules, followed by the formation of actin stress fibers. At higher KGP18 concentrations, activated HUVECs rounded, contracted, formed cellular blebs and detached from the gelatin substratum. These morphological changes were similar to those observed with CA4 and OXi8006 [17] and were also observed with KGP18-treated MDA-MB-231 breast cancer cells (Appendix A).

### 3.2. KGP265 Dose Response in Orthotopic MDA-MB-231 Tumors in Nude Mice

Bioluminescence imaging (BLI) showed intense light emission from MDA-MB-231-luc tumors following the subcutaneous administration of luciferin to nude mice, reaching a maximum intensity 10–15 min post-administration, followed by a gradual decline over the next 15 min (Figure 3). For individual control tumors, the BLI curves were highly reproducible (Figure 3A), although a significant increase in the signal was observed for the group of control tumors (*n* = 5) over 24 h, attributable to tumor growth (*p* < 0.05; Figure 3G). Following a dose of KGP265 (5 mg/kg), a significantly lower light emission was observed following the administration of fresh luciferin two hours later (*p* < 0.05; Figure 3B,G), which remained significantly depressed at 4 h but showed considerable recovery by 24 h. At higher doses, the response was more significant at 2 h (*p* < 0.001), with little, if any, recovery at 24 h (Figure 3C–G). The response at 4 h was significantly greater than 2 h for 30 mg/kg KGP265, and the responses appeared to last longer (Figure 3G). Over a period of 24 h, both 25 and 30 mg/kg KGP265 caused significantly greater signal loss than 5 mg/kg KGP265 (*p* < 0.05). At the 4 h time point, all doses of KGP265 resulted in significantly less light than saline (*p* < 0.01) and doses of ≥7.5 mg/kg KGP265 resulted in lower light emission compared to 5 mg/kg KGP265 (*p* < 0.05). For individual doses of KGP265, the light emission was significantly depressed at 2, 4 and 24 h for 7.5, 10, 25 and 30 mg/kg KGP265 (*p* < 0.001), while, for 5 mg/kg KGP265 (*p* < 0.05), the depression was only significant at 2 and 4 h. For the saline control, the signal increased significantly between 4 and 24 h (*p* < 0.01), all based on Fisher’s PLSD. At doses exceeding 10 mg/kg KGP265, some mice died; meanwhile, there was extensive vascular recovery by 24 h after 5 mg/kg KGP265, and therefore, we chose a dose of 7.5 mg/kg for the further imaging investigations.

The BLI showed a highly reproducible signal from 4T1-luc tumors (Figure 4A,B), which was significantly diminished within 2 h of administering (IP) 7.5 mg/kg KGP265 (>85% signal decrease (*p* < 0.001), Figure 4C,D). In many cases, the tumor appeared black within 2 h (Figure 4D), and the tumor surface indicated a crater-like scar after 24 h (Figure 4E). MSOT showed a distinct spatial heterogeneity of vascular oxygen saturation (sO_2_) in 4T1-luc tumors at the baseline while breathing air and a range of small regional responses to the oxygen gas breathing challenge, which was highly consistent over 24 h (Figure 4F,H). Meanwhile, sO_2_ and the response to the oxygen gas breathing challenge (ΔsO_2_) were significantly depressed within 2 h of the KGP265 treatment and remained so after 24 h (*p* < 0.001; Figure 4G,I). The ΔsO_2_ response of the spine showed no significant change following treatment with KGP265 (*p* > 0.05).

A very similar behavior was observed in the RENCA-luc tumors (Figure 5). The BLI signal was significantly reduced within 2 h of 7.5 mg/kg KGP265. The ΔsO_2_ observed in RENCA-luc tumor was also severely depressed. We noted that the RENCA-luc tumors were considerably larger than 4T1-luc (*p* < 0.001; Figure 5E,G vs. Figure 4F,H).

To further evaluate the OE-MSOT image data, we defined a change in the vascular response
ΔΔsO_2_ = ΔsO_2_^Treatment^ − ΔsO_2_^Baseline^(4)
indicative of vascular development or impairment, as shown for each tumor type with respect to KGP265 after 2 and 24 h (Figure 6A,B). Compared to the untreated group, KGP265 caused significantly different ΔΔsO_2_. In concert with a diminished sO_2_ response, the BLI signal was depressed at 2 h and 24 h in both the 4T1-luc and RENCA-luc tumors (*p* < 0.001; Figure 6A,B).

Comparative histology with H&E staining showed substantially increased necrosis and hemorrhage in tumors 24 h after the treatment with KGP265 (Figure 7A and Figure 8A). Pimonidazole staining was significantly increased 24 h after the administration of KGP265, implying that the tumor vasculature was shut down in 4T1 tumors, causing increased hypoxia and consequent necrosis. Ki67 was significantly reduced following the KGP265 treatment (Figure 7). A very similar behavior was observed in the RENCA-luc tumors (Figure 8), and they showed extensive hemorrhage and multiple necrotic cores.

### 3.3. Comparison of Tumor Growth Delay for Therapeutic Cohorts

The twice-weekly dosing of 4T1-luc tumor-bearing mice with KGP265 alone (3 mg/kg) caused a modest but significant tumor growth delay (*p* = 0.002; Figure 9A). Low-dose carboplatin alone (30 mg/kg) caused a minimal tumor growth delay, with 5.8-fold growth over 14 days as compared to 6-fold for saline (*p* = 0.069). The low-dose combined therapy caused a significant tumor growth delay over 14 days compared to those receiving saline (*p* < 0.001), and the tumors were significantly smaller than those receiving low-dose carboplatin alone (*p* < 0.001). None of the groups showed weight loss (Figure 9C). the actual survival to the end point of 4T1 mice receiving low-dose combined therapy was significantly better than any other group (log-rank *p* = 0.007; Figure 9E).

As expected, the control MDA-MB-231-luc tumors showed an approximately 4-day volume doubling time (VDT) with 5.6-fold growth over 14 days (Figure 9B). At 14 days (after a fourth round of therapy), the MDA-MB-231-luc tumors on mice receiving the high-dose combination therapy (5 mg/kg KGP265 + 50 mg/kg carboplatin) were significantly smaller than those receiving carboplatin (50 mg/kg) or KGP265 (5 mg/kg) (*p* < 0.001) alone, while the tumors associated with all the treatment groups were significantly smaller than those receiving saline (*p* < 0.001). The low-dose combined therapy (3 mg/kg KGP265 + 30 mg/kg carboplatin) indicated 3.8-fold growth over 14 days compared to 5.6-fold growth for saline (*p* < 0.001). The combined high-dose therapy indicated only a 35% increase in volume over 14 days, but at this stage, the high-dose combined therapy group mice were sacrificed due to excessive weight loss. Indeed, low- or high-dose carboplatin caused significant weight loss (>10%), which was particularly severe in the combination groups, while KGP265 alone showed no obvious weight loss (Figure 9D). The low-dose combination therapy enhanced the survival of MDA-MB-231 tumors up to 20 days compared with saline, but those receiving the combined high-dose therapy showed significant toxicity and poorer survival than the others (Figure 9F). We also showed a histological analysis of KGP265 and the combined therapy with carboplatin in Appendix A; the results clearly demonstrated that the combined therapy group developed much more necrotic area than the control saline and KGP265 and carboplatin sole treatments.

## 4. Discussion

In vitro, we established that both KGP18 and its phosphate prodrug KGP265 showed antimitotic effects via a blockade at the G2/M phase of the cell cycle in MDA-MB-231 breast cancer cells (Figure 2A). Furthermore, KGP265 was readily dephosphorylated by alkaline phosphatase enzyme activity to generate active KGP18, which is both a VDA and an antimitotic agent. The KGP18 treatment of a pre-established HUVEC tube network resulted in cell rounding and clumping that disrupted the overall network (Figure 2B). Rapid and profound cytoskeletal changes were observed in the activated endothelial cells treated with 20 nM KGP18 (Figure 2C) that included disruption of the microtubules, followed by the formation of actin stress fibers. These morphological changes are similar to those observed for the CA4 mechanism [17] and were also observed with the KGP18-treated MDA-MB-231 breast cancer cells. These results prompted us to examine their activity against diverse tumor types in vivo, particularly regarding acute vascular shutdown and the therapeutic response.

KGP265 showed significant efficacy in terms of vascular disruption, achieving greater than 90% vascular shutdown in three orthotopic breast and kidney tumor types in mice within 2 h after the administration of 7.5 mg/kg IP. A substantial shutdown was maintained for at least 24 h, as observed by the dynamic BLI and MSOT. At this dose, there was no obvious acute systemic toxicity, but significant tumor necrosis was identified, accompanied by increased hypoxia and reduced proliferation activity in the surviving tumor regions. The twice-weekly dosing of KGP265 (3 mg/kg) caused a modest but significant tumor growth delay. The addition of KGP265 to carboplatin therapy caused a significantly enhanced tumor growth delay, but it was accompanied by a substantial toxicity, which was also observed for carboplatin alone.

A dynamic BLI effectively revealed a dose-response activity in the luciferase-expressing tumors (Figure 3, Figure 4 and Figure 5), as reported previously for diverse VDAs [5,53]. Repeat measurements showed excellent signal reproducibility for the control mice receiving saline, while the treated mice showed significantly reduced BLI signals within 2 h of administration of KGP265. The signal loss tended to increase at higher doses. For doses <10 mg/kg, some signal recovery was observed at 24 h, but shutdown increased at later time points for the higher doses. The BLI is particularly facile to implement, providing high-throughput observations with multiple mice observed simultaneously. MSOT is more complicated to implement, but it revealed the heterogeneity of tumor vascular oxygenation, which was temporally quite stable over 24 h. KGP265 led to a significantly reduced vascular oxygenation within 2 h, as well as a significantly smaller response to an oxygen gas breathing challenge (Figure 4I and Figure 5H). Depressed oxygenation continued at 24 h, while the spine muscle served as an effective control tissue, remaining stable at each time point. 

MSOT has several potential advantages: tumors do not need to be transfected with a luciferase reporter gene, no exogenous contrast agent is required and spatial and temporal heterogeneity are apparent, along with the potential effects on other organs. We favor the use of an oxygen gas breathing challenge, which reveals the vascular function, as recommended by Tomaszewski et al. [56] and applied by us to examine the effects of various therapeutic agents [52,53].

Vascular damage leading to hypoxia led to a significantly increased necrosis in the 4T1 (Figure 7A) and RENCA (Figure 8A) tumors. As expected, hypoxia was found to be significantly increased in the surviving tissues following treatment with KGP265, while the cellular proliferation was diminished (Figure 7 and Figure 8).

A bi-weekly dosing of 4T1-luc tumor-bearing mice with KGP265 at 3 mg/kg caused a significant tumor growth delay but no survival beyond 17 days (Figure 9). Low-dose carboplatin (30 mg/kg bi-weekly) caused no apparent tumor growth delay, but some mice survived for 20 days. Combining KGP265 + carboplatin significantly enhanced the tumor growth delay compared with carboplatin alone. The tumors did continue to grow somewhat with the combined therapy, but 50% of the mice survived to 20 days. Similarly, in the MDA-MB-231-luc tumors, the low-dose combination yielded a significant tumor growth delay but no apparent advantage over carboplatin alone. Lesser efficacy in the MDA-MB-231 tumors compared with 4T1 may be a consequence of less sensitivity to KGP265; in Figure 3, we noted an about 85% less BLI signal in MDA-MB-231-luc tumors 2 h after 7.5 mg/kg KGP265, whereas the loss was over 95% for the 4T1-luc tumors (Figure 4). In an attempt to enhance the overall therapeutic efficacy, we examined a higher dose of KGP265, since we found an acute dose response (Figure 3). KGP265 alone (5 mg/kg twice-weekly) appeared as effective as the low-dose carboplatin (Figure 9B, F). Applying increased carboplatin in the high-dose regimen (50 mg/kg carboplatin + 5 mg/kg KGP265) essentially halted the tumor growth over two weeks but was accompanied by unacceptable toxicity. A severe weight loss was observed, ultimately requiring sacrificing of the mice after two weeks. Examining high dose carboplatin alone indicated less, but still significant, weight loss and much less tumor growth delay. Since no weight loss was observed with the KGP265 alone, it motivates further investigations of alternative combination therapies without carboplatin.

We note that MDA-MB-231 is a well-investigated tumor model for therapeutic investigations, providing reproducible consistent growth and results [43,57]. The 4T1 tumor is syngeneic and highly aggressive and has been examined with respect to CA1P treatment previously [48]. We are not aware of previous studies of RENCA tumors with respect to tubulin binding vascular-disrupting agents, but they were examined with respect to the alternative vascular-disrupting approach based on DMXAA [49].

## 5. Conclusions

We demonstrated, for the first time, that the novel benzosuberene prodrug KGP265 is readily dephosphorylated to generate KGP18, which generates vascular damage in both human tumor xenografts and syngeneic tumors in mice. A clear dose response was observed with acute vascular shutdown within two hours at doses as low as 5 mg/kg. The bi-weekly administration of KGP265 alone generated significant tumor growth delay, but it is clear that combination therapy enhances the delay and will be required to achieve longer-term control. Indeed, a combination with carboplatin was found to increase the tumor growth delay. These results demonstrated the efficacy of this new vascular-disrupting agent (KGP265), encouraging further investigations with other therapeutic combinations. 

## Figures and Tables

**Figure 1 cancers-13-04769-f001:**
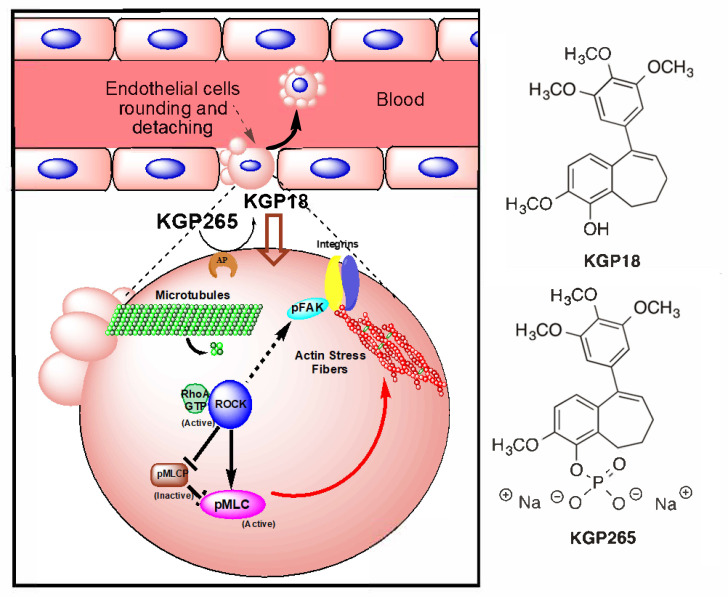
Proposed KGP265 mechanism in the tumor vasculature Rapid dephosphorylation of the prodrug KGP265 in the tumor vessel endothelium yields the active agent KGP18, which binds to tubulin, inhibits tubulin polymerization and leads to the activation of RhoA kinase (ROCK). The Phosphorylation of the downstream ROCK targets results in the formation of actin stress fibers. As actin stress fibers resolve, endothelial cells become round, form blebs and detach, culminating in vascular disruption. See Figure 2 and Appendix A.

**Figure 2 cancers-13-04769-f002:**
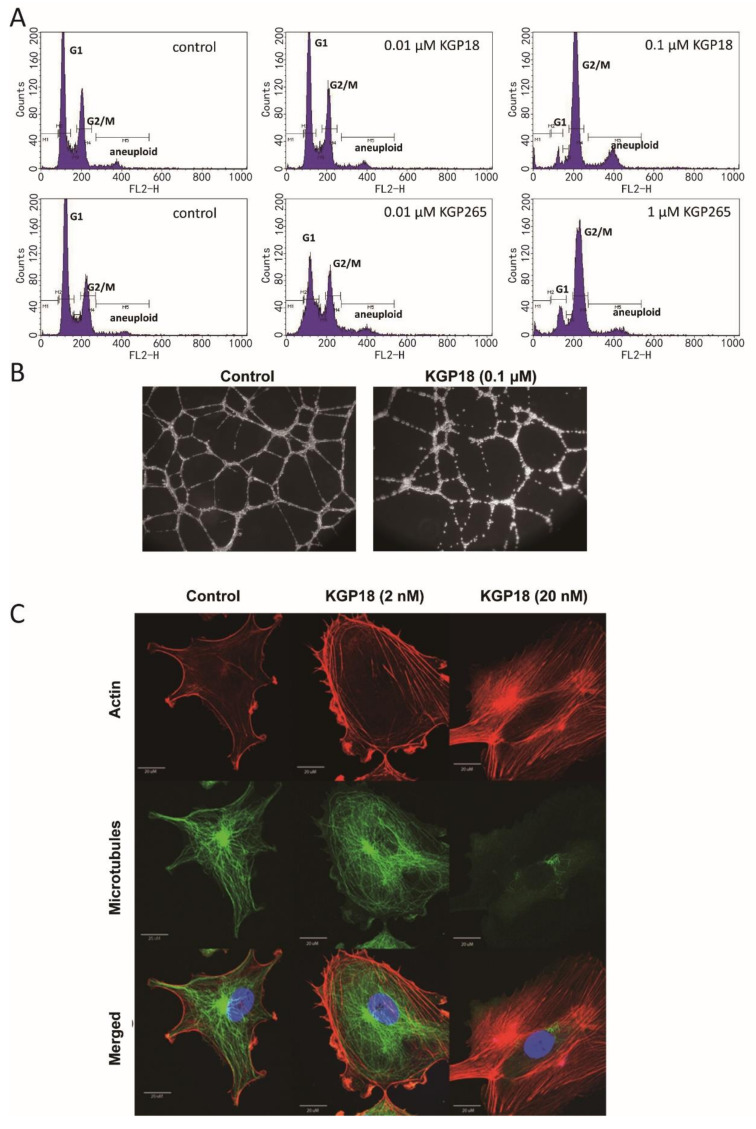
KGP18 treatment of MDA-MB-231 cells and activated HUVECs. (**A**) KGP18 (active agent) and the corresponding phosphate prodrug KGP265 induced concentration-dependent G2/M arrest in MDA-MB-231 cells, as assessed by flow cytometry. (**B**) KGP18 treatment disrupted capillary-like endothelial networks pre-established with HUVECs on Matrigel. (**C**) Monolayers of rapidly growing HUVECs underwent concentration-dependent changes in the cell morphology with KGP18 treatment (2 h) that demonstrated a loss of the microtubule structure and increased the bundling of filamentous actin into stress fibers. Representative confocal images of endothelial cells stained for α-tubulin (FITC), actin (Texas Red) and the nuclei (DAPI). Bar: 20 µm. HUVEC, human umbilical vein endothelial cell.

**Figure 3 cancers-13-04769-f003:**
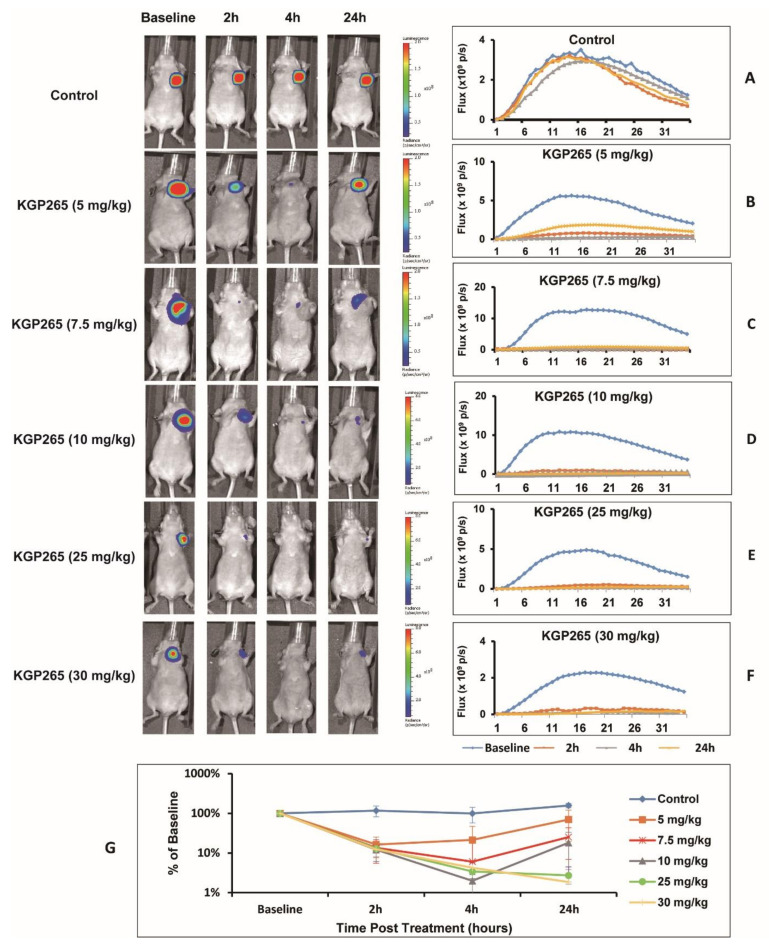
BLI observations of the MDA-MB-231-luc tumor dose response to KGP265. (**Left**) BLI signal intensity images overlaid as heat maps on gray-scale photographs of nude mice at about 12 min following the administration of luciferin at selected time points following the administration of various doses of KGP265. All intensity maps have the same heat scale. (**Right**) Corresponding BLI intensity curves for the respective individual mice on the left showing differential variations over a period of 35 min following the administration of luciferin at the baseline (blue), 2 h (orange), 4 h (grey) and 24 h (yellow) following each of the doses of KGP265: (**A**) saline control, (**B**) 5 mg/kg, (**C**) 7.5 mg/kg, (**D**) 10 mg/kg, (**E**) 25 mg/kg, (**F**) 30 mg/kg and (**G**) KGP265 dose response comparisons. Relative signals indicating differential dose responses over a period of 24 h following the administration of different doses of KGP265 to MDA-MB-231-luc tumor-bearing nude mice: saline control (*n* = 5, blue diamond), 5 mg/kg (*n* = 5, orange square), 7.5 mg/kg (*n* = 5, red asterisk), 10 mg/kg (*n* = 6, grey triangle), 25 mg/kg (*n* = 4, green circle) and 30 mg/kg (*n* = 6, yellow cross). Individual data points represent the mean ± SD.

**Figure 4 cancers-13-04769-f004:**
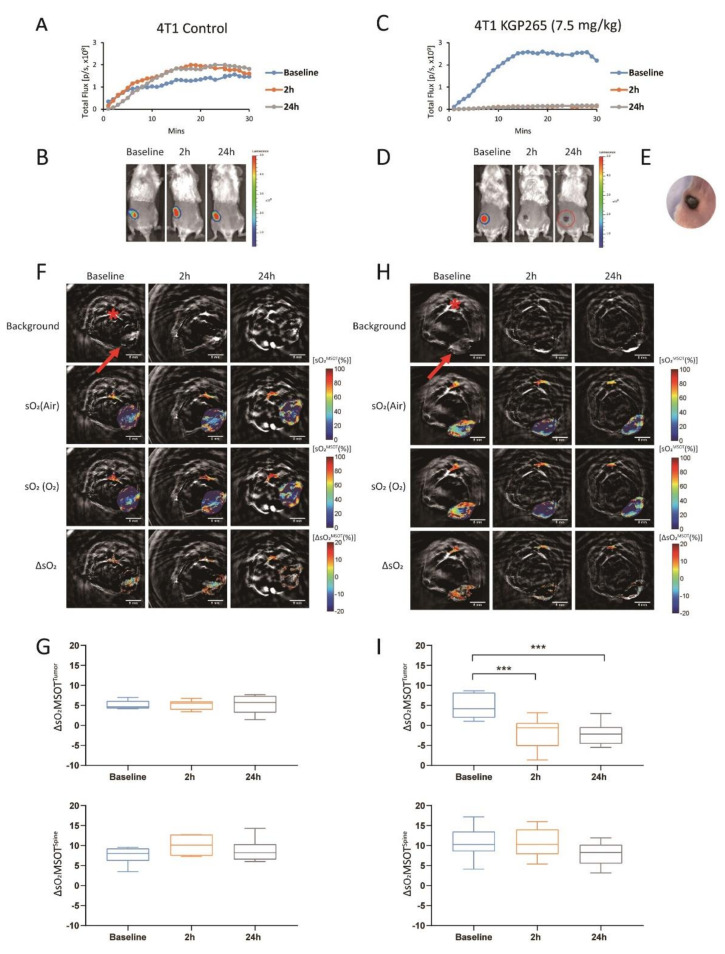
BLI and OE-MSOT of 4T1-luc mammary fat pad tumor response to KGP265. (**A**,**C**) BLI intensity curves for individual mice showing differential variations over a period of 30 min following the administration of luciferin at the baseline (blue), 2 h (orange) and 24 h (grey) following the administration of saline or KGP265 (7.5 mg/kg). (**B**,**D**) BLI signal intensity images overlaid as heat maps on gray-scale photographs of the BALB/c mice in (**A**,**C**), respectively, at about 20 min following the administration of luciferin at selected time points following the administration of saline or KGP265. All the intensity maps have same heat scale. (**E**) Expansion of the area indicated by the red circle in (**D**) indicating the region developing a black scar on the tumor surface over a period of 24 h following the administration of KGP265. (**F**,**H**) MSOT images showing cross-sections of 4T1 tumor-bearing mice with respect to saline or KGP265 (7.5 mg/kg): top single wavelength: 800-nm image showing anatomical details with parametric maps below. The spine ROI is indicated by the red star, and the tumor ROI is indicated by the red arrow. Spatial distribution of oxygen saturation when the mouse breathed air (sO_2_^MSOT^ (Air)) or during the oxygen challenge (sO_2_^MSOT^ (O_2_)) and the magnitude of the response (ΔsO_2_) for each time course from breathing the air to breathing oxygen. The change in sO_2_ (ΔsO_2_) was calculated by subtracting the average tumor sO_2_, while breathing air was calculated from the average tumor sO_2_ while breathing oxygen. (**G**,**I**) Quantification of ΔsO_2_ for each group at different time points. For the statistical analysis, the levels in the treated tumors were compared to the levels in the control tumors with a two-way unmatched ANOVA test. ***, *p* < 0.001. The box is between the 25th and 75th percentiles, the line is at the median.

**Figure 5 cancers-13-04769-f005:**
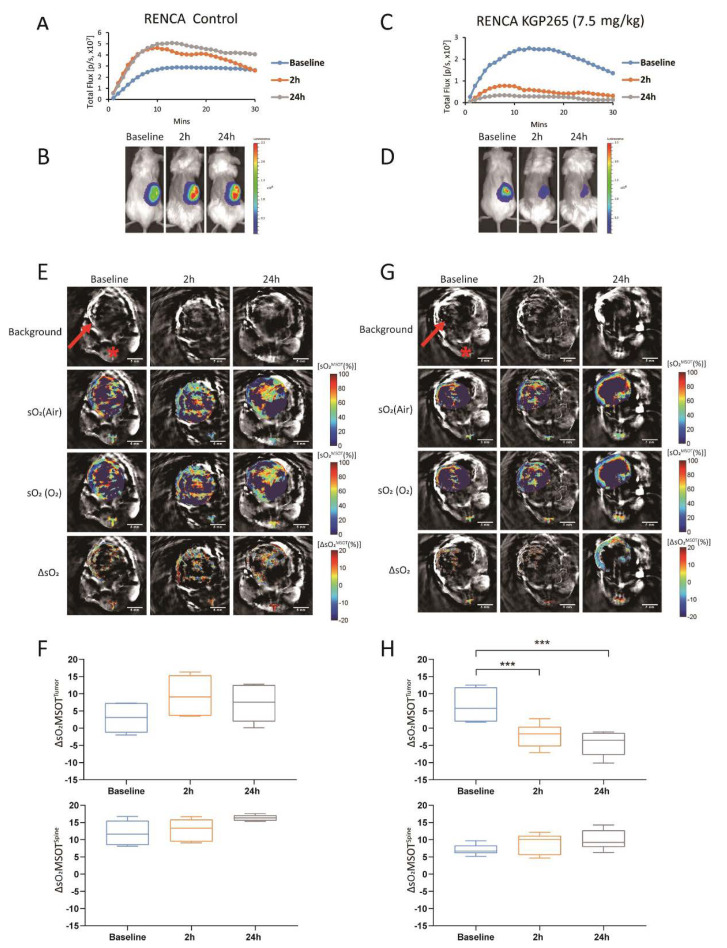
BLI and OE-MSOT of the RENCA-luc orthotopic kidney tumor response to KGP265. (**A**,**C**) BLI intensity curves for individual mice showing differential variations over a period of 30 min following the administration of luciferin at the baseline (blue), 2 h (orange) and 24 h (grey) following the administration of saline or KGP265 (7.5 mg/kg). (**B**,**D**) BLI signal intensity images overlaid as heat maps on gray-scale photographs of the BALB/c mice in (**A**,**C**), respectively, at about 10 min following the administration of luciferin at selected time points following the administration of saline or KGP265; all intensity maps have the same heat scale. (**E**,**G**) MSOT images showing cross-sections of RENCA tumor-bearing mice with respect to saline or KGP265 (7.5 mg/kg): background single wavelength (800 nm) image showing anatomical details with parametric maps below. The spine ROI is indicated by the red star, and the tumor ROI is indicated by the red arrow. Spatial distribution of the oxygen saturation when the mouse breathed air (sO_2_^MSOT^ (Air)) or during the oxygen challenge (sO_2_^MSOT^ (O_2_)), and the magnitude of the response (ΔsO_2_) for each time course from breathing air to breathing 100% oxygen. The change in sO_2_ (ΔsO_2_) was calculated by subtracting the average tumor sO_2_ while breathing air from the average tumor sO_2_ while breathing oxygen. (**F**,**H**) Quantification of the ΔsO_2_ for each group at different time points. For the statistical analysis, the levels in the treated tumors were compared to the levels in the control tumors with a two-way unmatched ANOVA test. ***, *p* < 0.001. The box is between the 25th and 75th percentiles, and the line is at the median.

**Figure 6 cancers-13-04769-f006:**
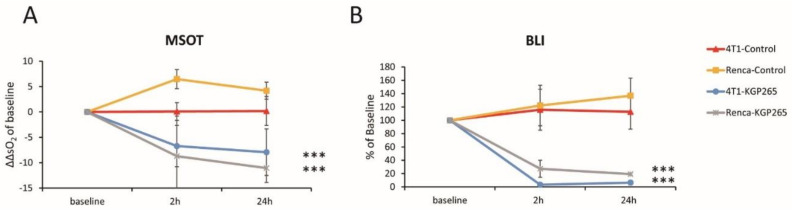
BLI and OE-MSOT responses to KGP265 were strongly correlated in different tumor types. (**A**) The tumor oxygen supply capacity is determined by the amplitude of the vascular response (ΔsO_2_) to the oxygen gas breathing challenge. A change in the ΔsO_2_ (ΔΔsO_2_) is indicative of vascular development or impairment and is shown for each tumor type with respect to the interventions at 2 and 24 h: RENCA tumors: saline control (yellow, *n* = 3) and KGP265 (grey *n* = 3) and 4T1 tumors: saline control (orange, *n* = 3) and KGP265 (blue, *n* = 4). Individual data points represent the mean ± SD. Fisher’s PLSD analysis of variance (ANOVA) indicated that 4T1 and RENCA tumors both showed a significantly reduced vascular response to the oxygen gas breathing challenge within 2 h (*p* < 0.0001) of the administration of KGP265, which remained depressed at 24 h (*p* < 0.001). ***, *p* < 0.001. (**B**) The corresponding relative BLI intensity changes observed for these tumors. Individual data points represent the mean ± SD. ANOVA based on Fisher’s PLSD indicated that the 4T1 and RENCA tumors both showed significantly reduced light emission within 2 h (*p* < 0.001), which remained significantly depressed at 24 h after the administration of KGP265 (*p* < 0.001).

**Figure 7 cancers-13-04769-f007:**
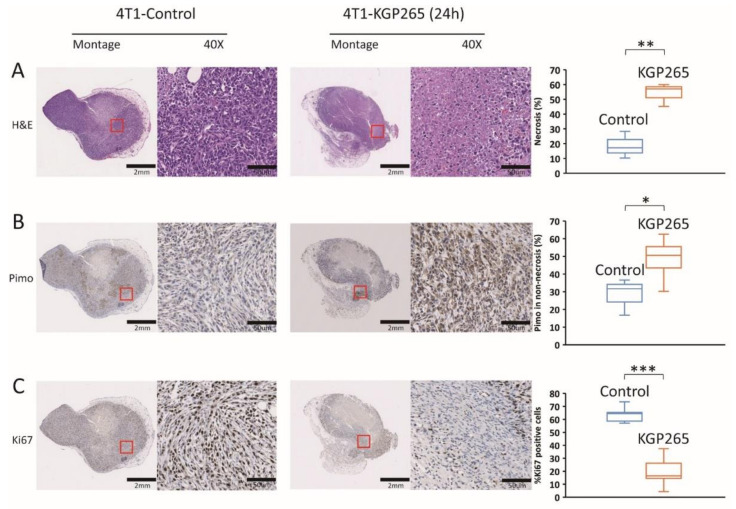
Histological analysis with IHC staining in 4T1 tumors in response to KGP265. (**A**) H&E staining showed differences between the control and KGP265 treated 4T1 tumor tissues. (**B**) IHC images of 4T1 tumor tissue sections, and graphs showing the levels of pimonidazole in the control and KGP265 treated tumors. (**C**) Representative IHC images of 4T1 tumor tissue sections, and graphs showing the levels of Ki67 in the control and KGP265-treated tumors. The red rectangles in the montages show expanded regions in 40× images. Scale bar: montage, 2 mm; 40×, 50 µm. The staining levels were quantified, and the data were plotted as the mean ± SEM. The values shown in the graphs are averages of the signals quantified from three independent tumors in the IHC experiments. For the statistical analysis, the levels in the treated tumors were compared to the levels in the control tumors with an unpaired *t*-test. * *p* < 0.05, ** *p* < 0.01 and *** *p* < 0.001.

**Figure 8 cancers-13-04769-f008:**
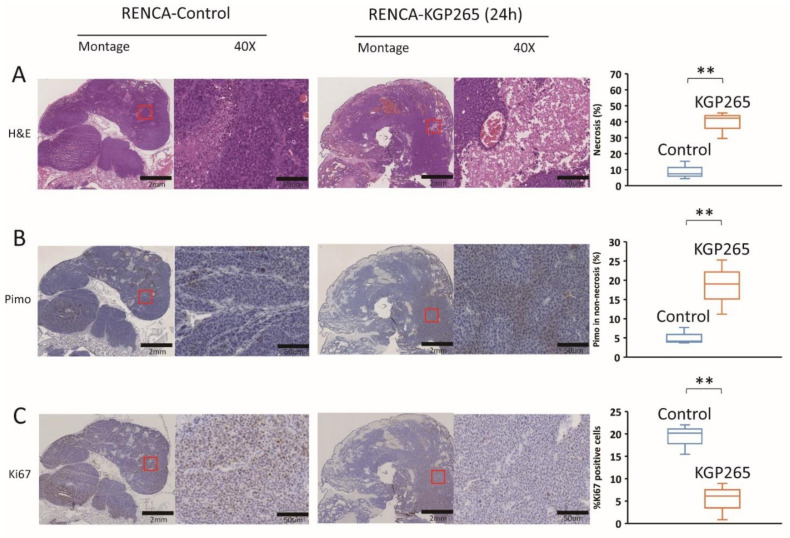
Histological analysis of IHC staining of RENCA tumors in response to KGP265. (**A**) H&E staining shows differences between the control and KGP265-treated RENCA tumor tissues. (**B**) IHC images of RENCA tumor tissue sections, and graphs showing the levels of pimonidazole in the control and KGP265-treated tumors. (**C**) Representative IHC images of RENCA tumor tissue sections, and graphs showing the levels of Ki67 in the control and KGP265-treated tumors. The red rectangles in the montages show the regions in 40× magnified images. Scale bar: montage, 2 mm; 40×, 50 µm. The staining levels were quantified, and the data are plotted as the mean ± SEM. The values shown in the graphs are averages of the signals quantified from three independent tumors in the IHC experiments. For the statistical analysis, the levels in the treated tumors were compared to the levels in the control tumors with an unpaired *t*-test. ** *p* < 0.01.

**Figure 9 cancers-13-04769-f009:**
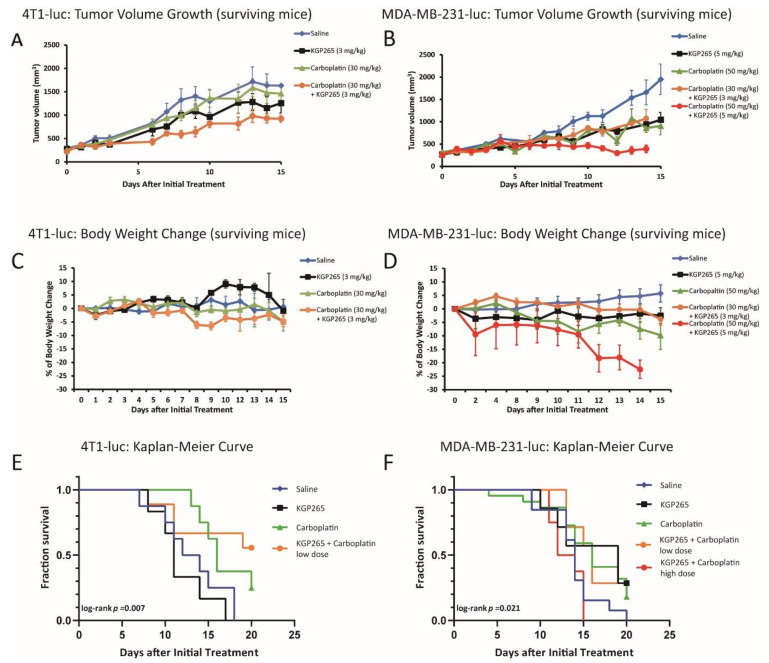
Comparison of the tumor growth curves, body weight loss and Kaplan-Meier survival curves for the therapeutic cohorts. (**A**) Growth curves for the groups of 4T1-luc tumors in the mammary fat pad of BALB/C mice: saline control (blue, *n* = 6), KGP265 (3 mg/kg) (black, *n* = 4), carboplatin (30 mg/kg) (green, *n* = 8) and carboplatin (30 mg/kg) + KGP265 (3 mg/kg) (orange, *n* = 6). All groups represented only mice surviving over the whole time course. The bars represent the SEM. (**B**) Growth curves for the groups of orthotopic MDA-MB-231-luc tumors growing in the mammary fat pad of NOD/SCID mice: saline control (blue, *n* = 13), KGP265 (5 mg/kg) (black, *n* = 7), carboplatin (50 mg/kg) (green, *n* = 22), carboplatin (30 mg/kg) + KGP265 (3 mg/kg) (orange, *n* = 8) and carboplatin (50 mg/kg) + KGP265 (5 mg/kg) (red, *n* = 8). All groups showed the surviving mice. The bars represent the SEM. (**C**) The mean change in the body weights for the cohorts of BALB/C mice bearing orthotopic 4T1-luc tumors in the mammary fat pad. All the agents were administered twice-weekly: saline control (blue), KGP265 (3 mg/kg) (black), carboplatin (30 mg/kg) (green) and carboplatin (30 mg/kg) + KGP265 (3 mg/kg) (orange). (**D**) The mean change in the body weights for the cohorts of NOD/SCID mice bearing orthotopic MDA-MB-231-luc tumors in the mammary fat pad. All the agents were administered twice-weekly: saline control (blue), KGP265 (5 mg/kg) (black), carboplatin (50 mg/kg) (green), carboplatin (30 mg/kg) + KGP265 (3 mg/kg) (orange) and carboplatin (50 mg/kg) + KGP265 (5 mg/kg) (red). (**E**) Kaplan-Meier survival curves for the groups of 4T1-luc tumors: saline control (blue), KGP265 (3 mg/kg) (black), carboplatin (30 mg/kg) (green) and carboplatin (30 mg/kg) + KGP265 (3 mg/kg) (orange). (**F**) Kaplan-Meier survival curves for the groups of MDA-MB-231-luc tumors: saline control (blue), KGP265 (3 mg/kg) (black), carboplatin (30 mg/kg) (green), carboplatin (30 mg/kg) + KGP265 (3 mg/kg) (orange) and carboplatin (50 mg/kg) + KGP265 (5 mg/kg) (red).

## Data Availability

We will make the original data available to investigators upon reasonable request.

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
