# Peer review of "Imaging-Guided Evaluation of the Novel Small-Molecule Benzosuberene Tubulin-Binding Agent KGP265 as a Potential Therapeutic Agent for Cancer Treatment"

_cancers, 2021, doi:10.3390/cancers13194769_

Round 1

Reviewer 1 Report

This manuscript assesses KPG treatment of breast cancer to shrink tumor and reduce blood vessel integrity. Overall, the manuscript is well written and assesses an important problem in the field of cancer treatment. The manuscript combines MSOT and bioluminescence imaging to assess tumor viability and blood oxygenation.  While I am very excited about this manuscript, there are some concerns that need to be addressed.

Major concerns:

  1. Why is the location of the tumor in Fig 3 different than in Fig 5?
  2. A great deal of the text in the figures is blurry.  This could be due to the PDF conversion, but this needs to be addressed.
  3. The majority of the dynamic range scale bars in the manuscript, particularly all bioluminescence images, are too small and prevent the reader from analyzing the data.
  4. The majority of the images in the manuscript are too small and there are too many panels/figure with text that is generally too small to read.
  5. The gray-scale background image 5E and 5G need to be lighter as the high levels of black impede the ability of the reader from seeing the deoxygenated areas (in blue/purple).  Arrows need to be placed at the areas of focus to allow the reader sufficient orientation.  These images are also a little confusing as the same color scale bar goes from 100 to 0 in some panels and from 20 to -20 in others.  If these ranges are truly different, then a different color scale bar should be used to emphasize the differences.
  6. Figure 3 needs to be simplified with the actual name/group included in the image.  Using labels (A-M labels) is confusing. Again, the scale bar is not readable.
  7. The micrographs in figure 6 need to have higher magnification (at least 40X, but 60-100X would be better) instead of 20X shown. It is not possible to analyze the H&E or Ki67 and Pimonidazole brown staining at 20X.  At present, these images prevent analysis and must improve.  Only due to the graphs in figure 6, the data can be analyzed.
  8. Some of the materials and methods needs to include further detail, i.e. the brand of the confocal microscope is missing, size of the region of interest is missing, was the region of interest measurement performed on a single slice or for the tumor as a whole, etc.

Author Response

This manuscript assesses KPG treatment of breast cancer to shrink tumor and reduce blood vessel integrity. Overall, the manuscript is well written and assesses an important problem in the field of cancer treatment. The manuscript combines MSOT and bioluminescence imaging to assess tumor viability and blood oxygenation.  While I am very excited about this manuscript, there are some concerns that need to be addressed.

Major concerns:

  1. Why is the location of the tumor in Fig 3 different than in Fig 5?

All tumors were implanted orthotopically in breast (MDA-MB-231-luc and 4T1-luc) or kidney (RENCA-luc) respectively. In our first tests, we used BLI only and implanted MDA-MB-231-luc breast tumors (Fig 3) in the left upper mammary fat pad of nude mice, but this location proved to be less satisfactory for MSOT due to motion. Thus for the combined BLI and MSOT evaluation we implanted 4T1-luc in the inguinal mammary fat pad (Fig 4). RENCA-luc was implanted in the right kidney to minimize potential signal interference from the very highly vascularized spleen (Fig. 5).

2. A great deal of the text in the figures is blurry.  This could be due to the PDF conversion, but this needs to be addressed.

We now provide jpeg files to enhance resolution and have uploaded figures separately.

3. The majority of the dynamic range scale bars in the manuscript, particularly all bioluminescence images, are too small and prevent the reader from analyzing the data.

We have now enlarged the BLI image range scale bars, as requested.

4. The majority of the images in the manuscript are too small and there are too many panels/figure with text that is generally too small to read.

We have enlarged text or whole figures, as suggested.

5. The gray-scale background image 5E and 5G need to be lighter as the high levels of black impede the ability of the reader from seeing the deoxygenated areas (in blue/purple).  Arrows need to be placed at the areas of focus to allow the reader sufficient orientation.  These images are also a little confusing as the same color scale bar goes from 100 to 0 in some panels and from 20 to -20 in others.  If these ranges are truly different, then a different color scale bar should be used to emphasize the differences.

We appreciate the suggestion, but we find that making the background lighter reduces the discrimination of the anatomical structures. We have added Arrows to assist the reader in identifying tissues. In terms of heat scales, it should be clear that the SO2 (air) and SO2 (O2) images both use 0 to 100%, whereas the delta SO2 is -20 to 20%. We have used such presentation in previous reports (ref 53). It is not clear that using a different color scheme would enhance clarity and we respectfully decline to do so.

6. Figure 3 needs to be simplified with the actual name/group included in the image.  Using labels (A-M labels) is confusing. Again, the scale bar is not readable.

We have modified the labeling as suggested and enlarged the range scale bars.

7. The micrographs in figure 6 need to have higher magnification (at least 40X, but 60-100X would be better) instead of 20X shown. It is not possible to analyze the H&E or Ki67 and Pimonidazole brown staining at 20X.  At present, these images prevent analysis and must improve.  Only due to the graphs in figure 6, the data can be analyzed.

As recommended we have replaced the 20X images by 40X (in Figure 7).

8. Some of the materials and methods needs to include further detail, i.e. the brand of the confocal microscope is missing, size of the region of interest is missing, was the region of interest measurement performed on a single slice or for the tumor as a whole, etc.

Details of microscopes are presented. Fluorescence and phase contrast images were collected using an Olympus FV 1000 con-focal microscope with Olympus fluoview software (Olympus Imaging America Inc., Center Valley, PA) using a 60x oil immersion objective. The measurements were presented on a single slice with region of interest covering the whole tumor in that slice. The selected field in the tumor periphery was imaged using a Hamamatsu NanoZoomer 2.0HT microscope (Hamamatsu, Japan); the setting is 20X stackable scan

Reviewer 2 Report

I could clearly understand the effect of a novel vascular disrupting agent, KGP265, in breast and kidney tumors using MSOT. I think that you could add BLI and MSOT data in combined therapy to clearly support your study.

I leave some comments. 
Abstract: Please provide in the results section the actual key points from the findings.

Methods: Please include about cell culture part.

Results: Please include Histological analysis, BLI, and MSOT data in combined therapy (Fig.9). There needs to be a conclusion

Author Response

Comments and Suggestions for Authors
I could clearly understand the effect of a novel vascular disrupting agent, KGP265, in breast and kidney
tumors using MSOT. I think that you could add BLI and MSOT data in combined therapy to clearly
support your study.
I leave some comments.
Abstract: Please provide in the results section the actual key points from the findings.
Given the 200 word limit, it is difficult to expand the results. We believe the salient points are presented.
Methods: Please include about cell culture part.
Details have been added, as requested. (page 5, lines 177-183 in revised manuscript)
Results: Please include Histological analysis, BLI, and MSOT data in combined therapy (Fig.9). There
needs to be a conclusion.
We added histological analysis of combined therapy of MDA-MB-231 tumor group as Figure S3.

Reviewer 3 Report

In this manuscript, the author perform the prodrug KGP265 have the potential to inhibit the TNBC breast cancer. They also show the significant combination effect with the carboplatin, and use the xenograft model to prove the low concentration used of KGP265. There have some questions hope the author can revise before the paper to be accepted.

  1. Can the author show the PK analysis of the KGP265?
  2. In this manuscript, the author does not clear show how much mouse use and died in each dose, the author should show the whole data to let reader can realize this compound.
  3. From the results of figure9 A and B, the KGP265 seems work well on 4T1-luc than MDA-MB-231 by comparing with the dose of 30mg/kg carboplatin. Can the author add some explanations of why this will happen in the manuscript?

Author Response

In this manuscript, the author perform the prodrug KGP265 have the potential to inhibit the TNBC breast cancer. They also show the significant combination effect with the carboplatin, and use the xenograft model to prove the low concentration used of KGP265. There have some questions hope the author can revise before the paper to be accepted.

  1. Can the author show the PK analysis of the KGP265?

To date we have not undertaken PK analysis of KGP265 and cannot perform in the time frame for resubmission. We do plan to do PK study in the future, very good suggestion.

2. In this manuscript, the author does not clear show how much mouse use and died in each dose, the author should show the whole data to let reader can realize this compound.

We use the survival mice for analysis and mention the numbers in fig 3 and 9 legend. For dose studies (fig 3): saline control (n=5, blue ♦), 5 mg/kg (n=5, orange ■), 7.5 mg/kg (n=5, red *), 10 mg/kg (n=6, grey▲), 25 mg/kg (n=4, green ●), 30 mg/kg (n=6, yellow x).

For fig.9: saline control (n=5, blue ♦), 5 mg/kg (n=5, orange ■), 7.5 mg/kg (n=5, red *), 10 mg/kg (n=6, grey▲), 25 mg/kg (n=4, green ●), 30 mg/kg (n=6, yellow x).

3. From the results of figure 9A and B, the KGP265 seems work well on 4T1-luc than MDA-MB-231 by comparing with the dose of 30mg/kg carboplatin. Can the author add some explanations of why this will happen in the manuscript?

As shown in figure 3, MDA-MB-231-luc tumor BLI signal was significantly diminished within 2 hrs of administering (IP) 7.5 mg/kg KGP265 (>85% signal decrease (p<0.001), meanwhile in Fig 4A  BLI signal  in 4T1-luc tumors was diminished >95%  (p<0.0001) within 2hrs of administering (IP) 7.5 mg/kg KGP265. This indicates that the 4T1 vasculature is more sensitive to VDA activity and may be the reason why KGP265 was more effective on 4T1-luc than MDA-MB-231 in the low dose combination studies with carboplatin, as now discussed.